# Anticytokine Autoantibodies and Fungal Infections

**DOI:** 10.3390/jof9080782

**Published:** 2023-07-25

**Authors:** Shanthi Kappagoda, Stanley Deresinski

**Affiliations:** Division of Infectious Diseases and Geographic Medicine, Stanford University, Stanford, CA 94304, USA; polishmd@stanford.edu

**Keywords:** anticytokine autoantibodies, talaromycosis, cryptococcosis, histoplasmosis, chronic mucocutaneous candidiasis, anti-interferon-γ antibodies, anti-GM-CSF antibodies, anti-IL-17/22 antibodies

## Abstract

Anticytokine autoantibodies (ACAAs) can cause adult onset immunodeficiencies which mimic primary immunodeficiencies and can present as refractory and severe fungal infections. This paper provides an overview of the role of innate immunity, including key cytokines, in fungal infections and then describes four clinical scenarios where ACAAs are associated with severe presentations of a fungal infection: (1) *Talaromyces marneffei* infection and anti-interferon-γ, (2) histoplasmosis and anti-interferon-γ, (3) *Cryptococcus gattii* infection and anti-GM-CSF, and (4) mucocutaneous candidiasis and anti-IL-17A/F (IL-22). Testing for ACAAs and potential therapeutic options are discussed.

## 1. Introduction

Fungal infections can be superficial (for example involving skin, nails, or hair), cutaneous or mucosal, or invasive. Many medically important fungi only cause invasive disease in the setting of impaired immunity. Traditional risk factors include HIV, diabetes, malignancy, chemotherapy, and iatrogenic immunosuppression [1]. The investigation of patients who present with persistent, recurrent, or severe invasive fungal infections in the absence of traditional risk factors has uncovered immunological pathways involved in the defense against fungal pathogens and elucidated underlying primary immunodeficiencies or inborn errors of immunity [2]. In the last 20 years, investigations into patients with similar clinical presentations but with no known inborn error of immunity has led to the discovery of anticytokine antibodies (ACAAs), which convey unique susceptibilities to certain fungal infections. Here, we present four clinical scenarios where ACAAs are associated with severe presentations of a fungal infection: (1) *Talaromyces marneffei* infection and anti-interferon-γ, (2) histoplasmosis and anti-interferon-γ, (3) *Cryptococcus gattii* infection and anti-GM-CSF, and (4) mucocutaneous candidiasis and anti-IL-17A/F (IL-22). Testing for ACAAs and potential therapeutic options are discussed.

## 2. Role of Innate Immunity in Fungal Infections

The first lines of defense against invasive fungi are intact skin or mucosal epithelial surfaces and, for inhaled fungal pathogens, the mucociliary escalator. Organisms that evade these defenses encounter the major effector cells of the innate immune response—neutrophils, macrophages, and dendritic cells. The elimination of the fungi can occur either intracellularly after phagocytosis or extracellularly. An example of extracellular defense is the creation by neutrophils of extracellular traps, net-like structures comprised of DNA-histone complexes and proteins released by activated neutrophils, which are capable of capturing fungal forms too large for phagocytosis.

The process of phagocytosis begins when immune receptors called host pattern recognition receptors (PRRs) recognize pathogen associated molecular patterns (PAMPS) in the fungal cell wall, such as glucans, chitin, and mannoproteins. Critical PRRs expressed by innate immune system cells include Toll-like and C-type lectin receptors, such as dectin. The resultant activation of signaling pathways ultimately leads to NF-kB (nuclear factor kappa-light-chain-enhancer of activated B cells) and inflammasome assembly. This, along with phagocytosis results in oxidative and nonoxidative killing of the pathogen.

Upon phagocytosis of a fungal pathogen, macrophages initiate a chain of events (phagosome-lysosome fusion and the generation of reactive oxygen intermediates and reactive nitrogen intermediates) that destroy the pathogen. Dendritic cells are less efficient at killing engulfed pathogens, but like macrophages, they present antigens to the T cells, thus engaging the adaptive immune response.

Shortly after resident neutrophils, macrophages, and dendritic cells encounter fungal antigens, they release multiple cytokines, which recruit additional effector cells of the innate immune system, activate the inflammatory response, and stimulate the response of the adaptive immune system. We will focus here on three cytokines of clinical significance—granulocyte macrophage colony stimulating factor (GM-CSF), interleukin 17A (IL-17A) and interferon-gamma (IFN-γ). Autoantibodies that neutralize these cytokines are associated with specific clinical syndromes and heightened susceptibility to intracellular pathogens including some fungi. This can result in severe and refractory presentations of fungal infections.

GM-CSF is produced by multiple cell types, including fibroblasts and endothelial cells as well as neutrophils, macrophages, and dendritic cells [3]. It stimulates the maturation, differentiation, and activation of dendritic cells, granulocytes, monocytes, and macrophages, and augments mononuclear cell phagocytosis and antimicrobial activity. GM-CSF is required for terminal differentiation and function of alveolar macrophages, one function of which is to prevent pulmonary surfactant accumulation.

Of the six cytokines in the IL-17 family (IL-17A through F), IL-17A and F are produced by CD4 (specifically Th17) cells, among other cells of the adaptive immune system. IL-17, a pro-inflammatory mediator, stimulates the production of granulocyte colony-stimulating factor (G-CSF) and chemokines, such as CXCL1 and CXCL2. During fungal infections, IL-17 stimulates the production of antimicrobial peptides, such as defensin.

Lastly, IFN-γ is produced by CD4 (specifically naïve CD4 and Th1) cells, CD8 (specifically Tc1) cells, NK cells and macrophages. When IFN-γ binds to its receptor, it activates the Janus-activated kinase (JAK)-STAT pathway. It is critical to both innate and adaptive immunity as it activates macrophages, as well as stimulates natural killer cells and neutrophils.

## 3. Anticytokine Autoantibodies in Health and Disease

Anticytokine autoantibodies (including antibodies to IL-2, IL-8, TNF-α, and GCSF) may be found in low concentration in healthy individuals, blood donors, and in commercially available intravenous human immunoglobulin [4,5]. ACAAs are found in the circulation in immune complexes with their cytokine target, which are in equilibrium with free cytokine and free ACAAs [6]. They are hypothesized to play a regulatory role in our response to cytokines—for example by augmenting or diminishing cytokine signals or altering their half-life and thus modulating the inflammatory response [7,8]. Such presumptively regulatory antibodies in healthy people are generally present in low concentrations and do not appear to play a pathogenetic role. Their prevalence, however, increases with age, and autoantibodies to type I interferons have been associated with increased severity of influenza A and SARS-CoV-2 infections in elderly individuals [9,10].

Research into the anticytokine autoantibody profiles of patients with rheumatologic diseases, including systemic lupus erythematosus (SLE), rheumatoid arthritis, and Sjogren’s syndrome has demonstrated that the presence of different ACAAs may correlate with disease phenotype [11]. For example, autoantibodies to IFN-α are associated with decreased disease activity in SLE, and lower levels of anti-TNF-α antibodies are seen in disease exacerbations [6]. Subsequent research has shown that the dysregulation of cytokines mediating the responses of both the innate and adaptive immune system precede the development of clinical disease in SLE [6].

## 4. Autoantibodies in Non-Fungal Infections

The first reported case of anticytokine autoantibodies (ACAAs) associated with a severe presentation of an infectious disease was reported in 2004 in a patient with extrapulmonary non-tuberculous mycobacteria (NTM), who was found to have autoantibodies to IFN-γ [12]. Since then, ACAAs have become a well-recognized cause of acquired immunodeficiency. The clinical phenotype of patients with ACAAs can mimic that of patients with known primary immunodeficiencies or immune dysregulation caused by an inborn error of immunity in the same cytokine pathway [13]. This has led to the term ‘phenocopy’ to describe patients with ACAAs [7]. In contrast to patients with primary immunodeficiencies, patients with ACAAs present later in life.

The most well-characterized anticytokine antibody phenotypes that mimic primary immunodeficiencies are: (1) anti-IFN-γ, which is associated with disseminated NTM infections, reactivation of *varicella-zoster* virus, cytomegalovirus, *Salmonella* and *Toxoplasma* infections; (2) anti-GM-CSF, which is associated with disseminated *Cryptococcus gattii*, *Nocardia* and *Aspergillus* infections, as well as pulmonary alveolar proteinosis; (3) anti-IL-6, which is associated with increased susceptibility to staphylococcal infections; (4) anti-IFN-α, which is associated with autoimmune polyglandular syndrome-1 (APS-1), thymoma and severe COVID-19 infection; and (5) anti-IL-17/IL-22, which is associated with APS-1, thymoma, and chronic mucocutaneous candidiasis [7]. Figure 1 shows a visual representation of the diseases associated with different neutralizing anticytokine autoantibodies [8].

## 5. Autoantibodies in Fungal Infections

The remainder of this review will focus on four different fungal pathogens that can have severe presentations in association with the presence of specific ACAAs: (1) *Talaromyces marneffei* infection and anti-IFN-γ, (2) histoplasmosis and anti-IFN-γ, (3) *Cryptococcus gattii* infection and anti-GM-CSF, and (4) chronic mucocutaneous candidiasis and anti-IL-17A/F (IL-22).

## 6. *Talaromyces marneffei* and anti-IFN-γ Autoantibodies (AIGAs)

*Talaromyces* (formerly *Penicillium*) *marneffei* is a thermally dimorphic fungus, endemic primarily to tropical and subtropical Asia and Southeast Asia. The highest number of cases are reported from China, Thailand, and Vietnam. Infection is presumed to occur from inhalation of airborne conidia from soil.

After inhalation, the conidium transforms into its yeast form, which is taken up by macrophages residing in tissue. In most immunocompetent hosts, the organism is eliminated by the macrophages, and the infection is either asymptomatic or mild and self-limited. However, in hosts with impaired cell-mediated immunity, the yeast can proliferate in the macrophages and disseminate via the lymphatic system or hematogenously to cause severe infection. Clinical presentations of talaromycosis can range from a localized skin infection (due to inoculation) to pneumonia and disseminated disease. Disseminated disease can mimic tuberculosis, causing weight loss, fever, cough, lymphadenopathy, and hepatosplenomegaly. Patients with disseminated disease can also have skin and bone involvement [14].

In endemic areas, *T. marneffei* is primarily an opportunistic pathogen seen in people with HIV infection (typically with CD4 counts <100 cells/mm^3^), cancer, or undergoing immunosuppressive therapy [15,16]. It has been described in people with no known prior history of immunosuppression, although in these patients, preliminary laboratory findings were often suggestive of an undiagnosed immunodeficiency [17,18]. Further investigation of HIV-negative patients with disseminated *T. marneffei* infection (sometimes co-occurring with NTM infection) has led to the diagnosis of underlying immunodeficiency caused by AIGAs. For example, in a study of 58 patients with disseminated *T. marneffei* infection [19], AIGAs were detected in 55/58 cases (94%). These patients were also screened for the presence of autoantibodies to GM-CSF, IL-6, IL-17A, IL-12, and IL-23. Non-neutralizing GM-CSF autoantibodies were found in one patient, but no other autoantibodies were detected in this patient cohort or among 39 HIV-positive patients (19 of whom had *T. marneffei* infections). Among the 55 patients with AIGAs, common clinical features of *T. marneffei* infection included fever, cough, weight loss, and lymphadenopathy. All patients had pulmonary involvement, and many patients had disseminated disease, including lymph nodes (78%), skin (47%), bones and/or joints (24%), liver (14%), and spleen (9%).

Chen et al. [20] examined the cases of 42 HIV-negative patients with *T. marneffei* infection, of whom 20 were AIGA negative and 22 were AIGA positive. Compared to patients without AIGAs, patients who were AIGA positive were less likely to have underlying pulmonary disease, more commonly had systemic talaromycosis, and were more likely to have concurrent infections with other intracellular pathogens. Most patients who were AIGA-positive had poor outcomes despite receiving an appropriate antifungal treatment.

Research is ongoing into the pathogenesis of how AIGAs impair the control of *T. marneffei*. Interferon-γ plays a role in activating phagocytic cells to clear engulfed pathogens, such as *T. marneffei* and NTM. In patients with AIGAs, there is inhibition of the signal transducer and activator of transcription 1 (STAT1) phosphorylation and interleukin-12 production, resulting in the dysfunction of the Th1 response [14,21].

Chi et al. [22] have shown that AIGAs in adults with disseminated non-tuberculous mycobacteria (NTM) infections are associated with two specific HLA class II alleles: HLA-DRB1*16:02/DQB1*05:02 and HLA-DRB1*15:02/DQB1*05:01. HLA-DRB1*16:02/DQB1*05:02 is commonly found in populations in South China and Taiwan, whereas DRB1*15:02/DQB1*05:01 is more common in patients from Southeast Asia [23]. Interestingly, areas with a high prevalence of HLA-DRB1*16:02/DQB1*05:02 in South China are also areas that are endemic for *Talaromyces marneffei*.

Autoantibodies to IFN-γ target an epitope on the IFN-γ receptor that is critical for receptor activation. The amino acid sequence of this target epitope has a high degree of homology to a portion of the Noc2 protein of *Aspergillus* spp., resulting in cross-reactivity. This has led to the hypothesis that the generation of AIGAs may be an example of molecular mimicry resulting from exposure of genetically predisposed individuals to *Aspergillus* spp. [24].

## 7. Histoplasmosis and anti-IFN-γ Antibodies

Histoplasmosis in humans is caused by two varieties of the endemic dimorphic fungus *Histoplasma capsulatum—Histoplasma capsulatum var capsulatum*, which is found nearly worldwide, and *H. capsulatum var duboisii,* which is mostly found in West Africa. *H. capsulatum* is found in soil, especially in soil that has been enriched with bird or bat guano. Humans acquire infection through disruption of soil and inhalation of aerosolized microconidia. In East and Southeast Asia, histoplasmosis has been reported from China along the Yangtze river, central Myanmar, southern Thailand, northern Philippines, and parts of Indonesia [25]. As noted above, the discovery of adult onset immunodeficiency related to anticytokine autoantibodies initially focused primarily on HIV-negative patients with disseminated NTM infections. However, among patients with AIGAs in East Asia, fungal infections including cryptococcosis, talaromycosis and histoplasmosis are fairly common. For example, in one of the earliest investigations of the causes of adult onset-immunodeficiency in Thailand and Taiwan, patients with suspected immunodeficiency were recruited for a cohort study which included 52 people with disseminated NTM infections, 45 with another opportunistic infection (OI) with or without NTM coinfection, and 9 people with disseminated tuberculosis. Among the 45 people with NTM plus another OI, 43 (96%) had AIGAs and seven of the 45 (16%) had disseminated histoplasmosis [26].

This cohort was added to between 2010 and 2018 and reanalyzed, incorporating the previously described patients with AIGAs and adding additional patients, resulting in a total 97 patients with AIGAs—74 from Thailand and 23 from the United States. Among the 74 patients from Thailand, 10 (14%) had histoplasmosis on presentation [27].

These papers lack detailed clinical descriptions of patients with AIGAs who presented with histoplasmosis. There is one description of a patient with disseminated histoplasmosis who presented with fevers, weight loss, and an oral ulcer and was found to have necrotizing cervical lymphadenitis and a neck abscess. Blood, abscess, and fluid from a cervical lymph node biopsy grew *Histoplasma capsulatum* [28]. Subsequent investigations revealed an underlying immunodeficiency due to the presence of AIGAs [29].

## 8. *Cryptococcus gattii* Infection and anti-GM-CSF Autoantibodies

The invasive fungal infection cryptococcosis is caused by the encapsulated yeasts *Cryptococcus neoformans* and *Cryptococcus gattii*, although the taxonomy of these organisms is evolving as different strains are sequenced. Most human disease is caused by *C. neoformans,* which has an almost worldwide distribution. *Cryptococcus gattii* has a more restricted distribution, primarily in Western North America and tropical and subtropical regions, including Australia and Papua New Guinea [30]. Cryptococcosis is acquired most commonly via inhalation.

Autoimmune pulmonary alveolar proteinosis (aPAP) is a rare disease caused by an accumulation of surfactant in the alveoli, which leads to dyspnea and hypoxia. Patients with aPAP have high levels of anti-GM-CSF autoantibodies and, in addition to pulmonary insufficiency, are also susceptible to infections, such as disseminated nocardiosis, NTM, and fungal infections [3]. These infections may precede, occur concurrently, or succeed the development of pulmonary disease.

Interestingly, among Asian patients with no known immunodeficiency who present with cryptococcosis, a significant proportion have been found to have high levels of anti-GM-CSF neutralizing antibodies. For example, in a study of 39 patients from Taiwan with cryptococcal pneumonia, meningitis, or cryptococcemia, high titers of neutralizing anti-GM-CSF antibodies were found in 15/39 (38%) [31]. Fourteen of the 15 patients with anti-GM-CSF antibodies had a CNS disease. None of the 15 patients had a concomitant traditional risk factor (such as solid organ or hematopoietic stem cell transplant, hematologic malignancy, or immunosuppressive therapy). In a study published by the same group involving 23 apparently immunocompetent patients with cryptococcosis (9 with disseminated and 14 with localized disease), 3 of 12 patients with pulmonary disease and 1 of 2 patients with extrapulmonary infection had anti-GM-CSF antibodies detected [32]. In this study, the presence of anti-GM-CSF antibodies was not restricted to patients with CNS infection. Out of six patients with anti-GM-CSF antibodies and *C. gattii* infection (none with PAP), two had CNS and pulmonary involvement, three had pulmonary only, and one had a musculoskeletal infection.

The development of cryptococcal meningitis in the context of anti-GM-CSF antibodies seems to be relatively species-specific. In Wang et al. (2022) [31], 11 of the 14 patients with a CNS disease who were positive for anti-GM-CSF antibodies had confirmed *Cryptococcus gattii* infection, and 3 had *Cryptococcus* spp., whereas in the 12 patients with a CNS disease who were negative for anti-GM-CSF antibodies, 10 had *C. neoformans* and 2 had *Cryptococcus* spp. More data are needed to see if this apparent species-specificity prevails.

## 9. Chronic Mucocutaneous Candidiasis and anti-IL-17A/F (IL-22) Antibodies

*Candida albicans* is a part of the human skin and mucosal microbiome, but our innate and adaptive immune systems normally prevent it from causing invasive disease. Chronic mucocutaneous candidiasis (CMC) describes a syndrome that presents as unusually persistent, noninvasive *Candida albicans* infections of the skin, nails or mucosal surfaces that are resistant to topical therapy [33].

There are a variety of underlying immunodeficiencies that can present as CMC, but all of them are associated with impaired development of T helper 17 cells (Th17) leading to decreased IL-17 production or decreased IL-17 response [2]. Examples of primary immunodeficiencies, which are associated with CMC, are autosomal dominant hyper IgE syndrome (HIES, STAT3 deficiency), caspase recruitment domain family member 9 (CARD9) deficiency, and dectin-1 deficiency [7]. Th17 cells are a subset of CD4+ T helper cells (in addition to the Th1 and Th2 subsets), which play a role in defense against extracellular pathogens, especially on mucosal and epithelial surfaces. Th17 cells produce the cytokines IL-17A and IL-17F. These stimulate epithelial cells to express antimicrobial peptides and also increase neutrophil trafficking.

Not surprisingly, autoantibodies to Il-17A and IL-17F can also be associated with the CMC phenotype. However, when this occurs, it is typically in conjunction with an autoimmune polyendocrinopathy. Autoimmune polyglandular system Type 1 (APS-1), also known as autoimmune polyendocrinopathy candidiasis, ectodermal dystrophy (APECED) syndrome, is a rare autosomal recessive disease associated with CMC and autoimmune endocrinopathies, classically, hypoparathyroidism and adrenal insufficiency (Addison’s disease). The molecular mechanism underlying APECED is a lack of expression of a protein, autoimmune regulator (AIRE), in the thymus. AIRE enables thymic cells to express self-antigens. When an auto-reactive T cell binds to the self-antigen, the T cell undergoes apoptosis, removing this threat to self-tolerance. When there are defects in AIRE production, autoreactive T cells are not purged, leading to the autoimmune conditions seen in APECED, including vitiligo, keratosis, hepatitis, pancreatitis, type-1 diabetes mellitus, hypoparathyroidism, and adrenal insufficiency.

High titer antibodies to IL-17A, IL-17F, and/or IL-22 were found in 33 patients with APS-1, but none were found in 33 healthy controls, nor 103 patients with other autoimmune disorders [34]. In another study, Kisand et al. [35] found that the majority of the 162 patients with APECED had autoantibodies to IL-17A (41%), IL-17F (75%) or IL-22 (91%). However, not all patients with CMC have autoantibodies to these cytokines, and there are people with autoantibodies to these cytokines who do not have CMC, implying there are other mechanisms underlying this syndrome.

## 10. Diagnostic Considerations

Initial screening for the presence of AIGAs may be performed using the QuantiFERON-TB Gold^®^ test, an IFN-γ release assay, used to test for latent tuberculosis. The test is a functional assay, which uses ELISA to measure IFN-γ levels in whole blood samples in a negative control (nil) after exposure to a positive control (mitogen) and after exposure to tuberculosis specific antigens. For latent tuberculosis, a test is positive (i.e., indicative of IFN-γ release) if there is a response in the tuberculosis-specific antigen test relative to the nil tube. A test is classified as indeterminate if there is a failure to respond to the mitogen (positive control), which typically indicates anergy, or if there is a high background level of IFN-γ in the negative control tube.

The presence of AIGAs has been shown to inhibit detection of IFN-γ, resulting in low mitogen response in the positive control [36]. A larger study measured the plasma concentration of AIGAs, AIGA neutralizing capacity using ELISA and flow cytometry, and performed QuantiFERON-TB Gold In-tube (QFT-GIT) tests in 30 patients with disseminated NTM infection in Taiwan. This study found that among the 30 patients with neutralizing AIGAs, all 30 had indeterminate QFT-GIT results due to low or undetectable IFN-γ levels in the positive control (mitogen) tubes. This is in contrast to the four patients with a disseminated NTM infection who did not have AIGAs. In this patient group, none had an indeterminate QFT-GIT result, and their IFN-γ levels in the mitogen tube were significantly higher than the patients with AIGAs [37].

A positive screening test should be confirmed by more formal testing. The detection of AIGAs is most commonly made by ELISA, but the confirmation of neutralizing activity requires a functional assay, such as the inhibition of STAT1 phosphorylation, often achieved through flow cytometry [38]. The quantitation of AIGA levels may be helpful in assessing its relevance since low levels of antibodies are less likely to be relevant to the disease process. Measuring levels may also potentially be useful to follow the response to therapy. In the US, National Jewish has tests available for AIGAs.

Antibodies to GM-CSF may similarly be detected by ELISA with a subsequent functional assay, such as cell surface CD11b expression or quantification of intracellular phosphorylated STAT5 [3]. In the US, testing for anti-GM-CSF antibodies is available at National Jewish and Cincinnati Children’s Hospital.

Both ELISA and multiplex particle-based flow cytometry have been used to detect antibodies to IL-17 and IL-22 with confirmation of antibody specificity by Western blot [34]. Neutralization activity has been tested using SV-40-transformed fibroblasts stimulated with Il-17 followed by the measurement of IL-6 production in the supernatant using ELISA [34]. Commercial testing and functional assays for anti-IL-17 and IL-22 are not widely available in the US at this time.

## 11. Therapeutic Implications

Because antimicrobial therapy alone may often fail in the presence of anticytokine autoantibodies, in some patients, immunomodulatory therapy has been added to pathogen-directed therapy. Immunomodulatory therapy has included (1) efforts to eliminate autoantibodies, such as plasmapheresis or B-cell depletion (for example with rituximab) and (2) efforts to overcome or bypass the neutralizing effect of the autoantibodies (for example with exogenous GM-CSF administration) [7]. Phase III randomized controlled trial data for these therapies are lacking. While some Phase II studies have been conducted for patients with ACAAs—for example for rituximab and inhaled GM-CSF in autoimmune PAP—these studies have not included patients with infections. Data for use of immunomodulator therapy in patients with ACAAs and fungal infections are limited to case reports, which are subject to publication bias. Table 1 shows some potential immunomodulator therapies that have been considered for ACAAs.

Most data on immunomodulator therapy for ACAAs come from patients with NTM infections. Case report and case series data have been aggregated by Qiu et al. [48] in a systematic review of 810 cases of AIGA patients with infections, which included 79 from the authors’ records and 731 from the literature. Most of these patients did not have fungal infections (83% had NTM). Among these cases, 67/810 received immunomodulator therapy, 27/67 received rituximab, 22/67 cyclophosphamide, 8/67 cyclophosphamide plus glucocorticoids, 7/67 recombinant human IFN-γ, 4/67 rituximab plus corticosteroids, and 2/67 bortezomib. In addition, four patients were treated with one of the following: immunoglobulin, daratumumab, R-CHOP, and adalimumab plus glucocorticoids. In 48 of the 67 patients, AIGA titers decreased after immunotherapy. The duration of follow-up was highly variable among the included cases, but of these 67 patients, after AIGA immunotherapy, three patients died due to a worsening infection, and four had recurrent or persistent infection.

Zeng and colleagues [49] reported a case series of five AIGA-positive patients with disseminated talaromycosis who had experienced one or more relapses of infection after treatment with more than 6 months of antifungal therapy. Patients were treated with IV cyclophosphamide at 0.8–1.0 mg/m^2^ every 3–4 weeks for six cycles. Over 2 years of follow-up, it was noted that. AIGA titers fell after cyclophosphamide therapy. All five patients discontinued antifungal therapy after cyclophosphamide treatment; at 2 years, one patient had relapsed with *T. marneffei*, and four patients were considered cured.

The administration of exogenous IFN-γ is not expected to overcome the high titer of AIGAs reported in some cases of NTM infection [8]. The administration of GM-CSF has, however, been reported to improve pulmonary function in patients with autoimmune pulmonary alveolar proteinosis, although there is no evidence for its value in managing infections associated with anti-GM-CSF autoantibodies [3].

Most experience with the immunomodulator therapies listed in Table 1 has been in patients with mycobacterial infections and AIGAs; data on their use in patients with fungal infections are very sparse, making their potential benefit speculative at best. While immunotherapeutic approaches may intuitively make sense to target ACAA-mediated diseases, more clinical data are needed to define their role.

## 12. Conclusions

In patients without a previously known immunodeficiency or iatrogenic immunosuppression who present with severe or refractory fungal infections, anticytokine autoantibodies should be considered as a cause of underlying adult onset immunodeficiency. In particular, patients with disseminated *Talaromyces marneffei* infection or histoplasmosis should be screened for AIGAs, patients with *Cryptococcus gattii* infection for anti-GM-CSF antibodies, and patients with chronic mucocutaneous candidiasis for anti-IL-17A/F (IL-22), where possible.

Secondary immunodeficiencies caused by ACAAs are not common. Both this and the lack of awareness of ACAAs as an underlying cause of severe or refractory fungal infections are likely contributing to difficulty identifying patients with these conditions. In addition to the limited availability of testing in clinical laboratories, research on ACAAs has been hindered by a lack of standardization of the assays used for their detection, quantification, and function. The development of standardized criteria for interpreting titers and functional assays would help clinicians and researchers identify patients for study. Due their relative rarity, clinical therapeutic trials will require multicenter collaboration for patient recruitment.

## Figures and Tables

**Figure 1 jof-09-00782-f001:**
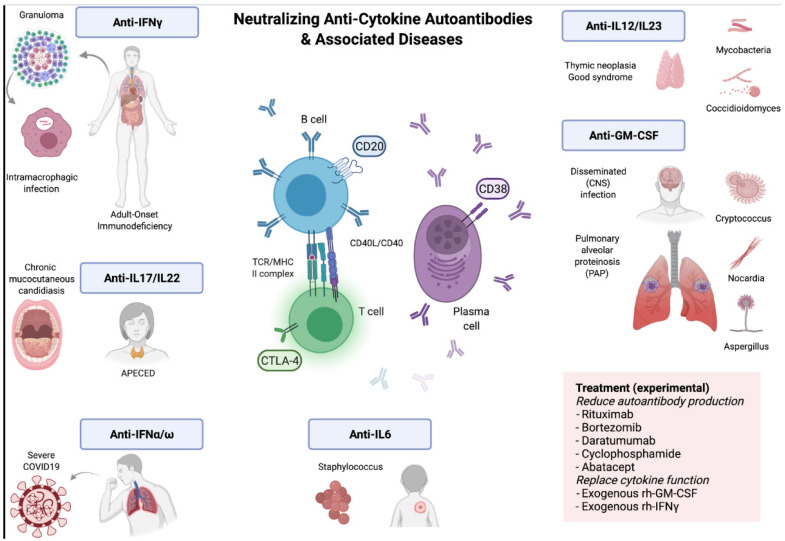
Reproduced with permission [8].

**Table 1 jof-09-00782-t001:** Potential immunotherapies for anticytokine autoantibodies.

Immunotherapy	Target	Mechanism	Citations
Rituximab	CD20	B cell depletion	[39] [40] [41] [42]
Daratumumab	CD38	Plasmablast, plasma cell, early B cell depletion	[43]
Abatacept	CD80 and CD86 on antigen-presenting cells	Block CD28 costimulation	[44]
Bortezomib	Proteasome	Plasma cell depletion	[45]
Cyclophosphamide	B cells	Impaired activation, proliferation, and differentiation	[46] [47] [40]

## Data Availability

No new data were created.

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
