# Peer review of "Anticytokine Autoantibodies and Fungal Infections"

_jof, 2023, doi:10.3390/jof9080782_

Round 1

Reviewer 1 Report

The manuscript prepared by Kappagoda et al. provides a comprehensive and in-depth review of the four most common anti-cytokine autoantibodies that lead to an increased susceptibility to invasive fungal infections. Overall, the manuscript presents several important existing knowledge in a well-structured and organized manner. However, there are a few noteworthy points that should be addressed to enhance the accuracy and clarity of the knowledge conveyed in this manuscript.

1. Figure 1: It is advisable to reconsider the inclusion of anti-IL12/23 autoantibodies in the figure due to the limited evidence and adoption regarding their association with coccidioidomycosis or mycobacterial infection.

2. Cryptococcosis and anti-GM-CSF autoantibodies: Regarding the spectrum of clinical manifestations associated with anti-GM-CSF autoantibodies, it is advisable to include some pertinent findings from the study by Kuo et al. (Clin Infect Dis 2022 Aug 25;75:278-287). For example, the authors reported that while previous research had established an association between anti-GM-CSF autoantibodies and severe cryptococcosis, often accompanied by central nervous system infections, cases where patients exclusively presented with pulmonary cryptococcosis were also observed. 

3. Diagnostic considerations: It will be valuable if the authors could provide an overview of the methods documented in the literature that can be utilized for identifying neutralizing anti-IL-17 and anti-IL-22 autoantibodies. 

4. Table 1: It could be misleading to incorporate all immunotherapies in the table without acknowledging their specific therapeutic potentials for different ACAAs. For instance, bortezomib has only demonstrated potential benefits as an adjunctive therapy in 2 patients with AIGA but not for the other ACAAs.

Author Response

Thank you for these comments which have helped us improve the quality of this submission. Please see the attached document - KappagodaDeresinskiResponsesReviewer1 with responses to to the issues raised. 

Reviewer 2 Report

The review discusses the role of anticytokine autoantibodies (ACAAs) in the context of fungal diseases. ACAAs are still very little explored in the literature and, therefore, the review does not bring as much new information that cannot be found in other reviews such as those focused on infectious diseases. Please see below some suggestions that can contribute to the improvement of this review:

- Please, the authors should add more information in the topic “3. Autoantibodies in non-fungal infections”. I believe that the factors that lead an individual to develop ACAAs should be better explained, in addition to pathogens, we know that therapeutic antibodies, growth factors, other biological agents and cytokines used in the treatment of  several diseases can also induce/overcome autoantibodies.

- Please explain further how ACAAs have a regulatory effect. The paper explores the bad side of ACAAs a lot, but their primary role is probably in regulation of the immune response. In autoimmune diseases such as rheumatoid arthritis or systemic lupus erythematosus, ACAAs have a good side since the increase in ACAAs is related to the decrease in clinical symptoms (role in the resolution of the disease). And from this initial explanation, explore how they contribute to the aggravation of the disease caused by external stimuli such as fungal infections.

- The amount of information about ACAAs in the literature is still limited. In conclusion, also add what, in your opinion, hinders research related to ACAAs. An example, such as the one cited, the diversity of techniques to detect them, which often makes understanding the prevalence and titer confusing, and mentioning what needs to be improved in research on, how important it is to identify whether these antibodies are really functional or not as this can influence the development of the disease and the course of the therapy, and etc.

Author Response

Thank you for these comments which have helped us improve the quality of our submission. Please see attached document which addresses the concerns raised. 

Round 2

Reviewer 2 Report

No more comments.